# Mitochondrial Localization Signal of Porcine Circovirus Type 2 Capsid Protein Plays a Critical Role in Cap-Induced Apoptosis

**DOI:** 10.3390/vetsci8110272

**Published:** 2021-11-10

**Authors:** Wanting Yu, Yuao Sun, Qing He, Chaoying Sun, Tian Dong, Luhua Zhang, Yang Zhan, Naidong Wang, Yi Yang, Yujie Sun

**Affiliations:** 1Biomedical Pioneer Innovation Center (BIOPIC), School of Life Sciences, Peking University, Beijing 100871, China; 1906390265@pku.edu.cn (W.Y.); yuao_sun@pku.edu.cn (Y.S.); Suncy@pku.edu.cn (C.S.); 2001110543@stu.pku.edu.cn (T.D.); 2School of Future Technology, Peking University, Beijing 100871, China; 3Hunan Provincial Key Laboratory of Protein Engineering in Animal Vaccines, Laboratory of Functional Proteomics (LFP), Research Center of Reverse Vaccinology (RCRV), College of Veterinary Medicine, Hunan Agricultural University, Changsha 410128, China; qinghe@stu.hunau.edu.cn (Q.H.); lhzhang@stu.hunau.edu.cn (L.Z.); yangzhan@hunau.edu.cn (Y.Z.); naidongwang@hunau.edu.cn (N.W.); yiyang@hunau.edu.cn (Y.Y.); 4Changde Research Center for Agricultural Biomacromolecule, College of Life and Environmental Sciences, Hunan University of Arts and Science, Changde 415000, China

**Keywords:** PCV2, Cap, MLS, subcellular localization

## Abstract

Porcine circovirus 2 (PCV2), considered one of the most globally important porcine pathogens, causes postweaning multisystemic wasting syndrome (PMWS). This virus is localized in the mitochondria in pigs with PMWS. Here, we identified, for the first time, a mitochondrial localization signal (MLS) in the PCV2 capsid protein (Cap) at the N-terminus. PK-15 cells showed colocalization of the MLS-EGFP fusion protein with mitochondria. Since the PCV2 Cap also contained a nuclear localization signal (NLS) that mediated entry into the nucleus, we inferred that the subcellular localization of the PCV2 Cap is inherently complex and dependent on the viral life cycle. Furthermore, we also determined that deletion of the MLS attenuated Cap-induced apoptosis. More importantly, the MLS was essential for PCV2 replication, as absence of the MLS resulted in failure of virus rescue from cells infected with infectious clone DNA. In conclusion, the MLS of the PCV2 Cap plays critical roles in Cap-induced apoptosis, and MLS deletion of Cap is lethal for virus rescue.

## 1. Introduction

Porcine circovirus 2 (PCV2) is the primary pathogen causing postweaning multisystemic wasting syndrome (PMWS), an emerging swine disease first discovered in Western Canada approximately 30 years ago [1,2,3]. PCV, the smallest virus to infect mammals, belongs to the family Circoviridae and the genus Circovirus. Four genotypes of PCV have been identified [1]. In addition to pathogenic PCV2, more recently, PCV3 and PCV4 have also been identified and are considered to be pathogenic, although PCV1 is not pathogenic [4,5]. Currently, PCV2 is one of the most important pathogens in the swine industry, due to its pathogenicity and continuous evolution [6].

PCV2 is a non-enveloped virus that packages a 1.7 kb single-stranded and circular DNA genome containing two major open reading frames (ORFs). ORF1 encodes two replication-associated proteins (Rep and Rep’) responsible for viral genome rolling loop replication in the nucleus [7,8,9]. ORF2 encodes a unique structural protein, the capsid protein (Cap), with 60 Cap subunits forming an icosahedral viral nucleocapsid that is ~17 nm in diameter [10,11]. Evidence from subcellular localization studies shows that the Cap allows PCV2 to shuttle between the cytoplasm and the nucleus during the infection cycle, due to the presence of a 41 amino acid nuclear localization signal (NLS) at its N-terminal end [12]. The Cap of PCV2 induced cell death and apoptosis in an epithelial cell line [13,14,15], indicating that it plays an important role in viral pathogenesis. More importantly, the Cap is also responsible for stimulating the production of neutralizing antibodies in pigs, and this effect has been exploited in vaccine design [16,17,18].

The main histological features that define PMWS are lymphocyte depletion and clustering of histiocytes and/or multinucleated giant cells in lymphoid tissues [2,19,20]. Electron microscopy studies have shown that the lymph nodes in pigs with PMWS have important ultrastructural changes, including severe hyperplasia and swelling of the mitochondria, as well as expansion and proliferation of the Golgi complex and rough endoplasmic reticulum [21]. Various intracytoplasmic inclusion bodies (ICIs) and intranuclear inclusion bodies (INIs) have been reported in macrophages, lymphocytes, hepatocytes, and epithelial cells, with PCV2 virus-like particles (VLPs) arranged in paracrystalline arrays [22]. ICIs were often detected adjacent to and inside mitochondria with severe pathological changes [23]. Specific immunogold labeling revealed the presence of a PCV2 antigen associated with the inner and outer mitochondrial membranes, and the mitochondrial antigen has been detected in ICIs [22]. These findings suggest that mitochondria play key roles in the PCV2 life cycle; however, the mechanisms by which viral particles are localized and transported to mitochondria are not well characterized.

In this study, we identified a mitochondrial localization signal (MLS) in the PCV2 Cap protein, which demonstrated the critical role of the newly identified MLS in Cap-induced apoptosis, and determined that infectious cloned DNA missing the MLS was incapable of virus rescue, demonstrating that MLS deletion of Cap is lethal for virus rescue.

## 2. Materials and Methods

### 2.1. Cell Culture

PK-15 cells free of PCV1, graciously provided by Nanjing Agricultural University, were maintained in high-glucose Dulbecco’s modified Eagle’s medium (DMEM) supplemented with l-glutamine (Life Technologies), 10% fetal bovine serum (FBS) (Life Technologies), 100 units/mL penicillin, and 100 mg/mL streptomycin (Life Technologies). The atmosphere was 5% CO_2_ and the incubator was maintained at 37 °C. For passaging, cells were digested with 0.25% trypsin (Life Technologies) and 10% FBS, after which they were collected by low-intensity centrifugation (<200× *g*).

### 2.2. Plasmid Construction

The pcDNA3.1-EGFP vector was purchased from Addgene, and the PCV2 *Cap* gene (GenBank accession number: MK281580) was synthesized by GenScript (Nanjing, China). The plasmid containing Mito-DsRed, which targeted mitochondria, was kindly provided by Dr. Yu Chen (Institute of Zoology, CAS, Beijing, China). The plasmids of Cap-EGFP, Cap(1–64)-EGFP, Cap(1–41)-EGFP, Cap(42–64)-EGFP, Cap(1–22)-EGFP, Cap(16–64)-EGFP and Cap(16–41)-EGFP were constructed by inserting Cap fragments of various lengths into pcDNA3.1-EGFP. The Cap plasmid was obtained by deleting EGFP from Cap-EGFP, and the Cap-DMLS (deletion of 16–41) plasmid was obtained by deletion of the MLS of Cap. PCV2 infectious clone DNA was pre-constructed in our laboratory [24], and the infectious clone DNA mutant of PCV2 was derived from it, with the MLS sequence deleted.

### 2.3. Cell Transfection

A transfection reagent (Chemifect; Fengrbio, Beijing, China) was used to transfect DNA into cells. Before transfection, Opti-MEME (Life Technologies, Carlsbad, CA, USA), a plasmid, and the transfection reagent were mixed in a specific ratio (200 µL:2 μg:3 μL) for 30 min. Meanwhile, cells were digested and seeded into the cell culture dishes at the appropriate density. The mixture was then added to the cells, and downstream experiments were performed after 24 h or 48 h.

### 2.4. Virus Rescue

After transfection with PCV2 infectious clone DNA and mutant DNA, PK15 cells were cultured and passaged for several generations. For each generation, half of the cells were retained for extraction of total DNA using a plasmid extraction kit (Tiangen, Beijing, China). To detect the viral genomic DNA in transfected and passaged cells, real-time PCR was performed using an ABI PCR system (Applied Biosystems, Waltham, MA, USA) with the extracted total DNA as the template. The primer sequences were as follows: pPCV2-Cap-qpcr-F: ctgttttcgaacgcagtgcc and pPCV2-Cap-qpcr-R: aactactcctcccgccatac.

### 2.5. Immunofluorescence Assay

Cells were fixed with 4% paraformaldehyde (PFA) (Solarbio, Beijing, China) for 15 min and washed three times with 1× PBS (Life Technologies, Carlsbad, CA, USA), for 5 min each time. Next, 5% BSA (Sigma-Aldrich, St. Louis, MO, USA) in 1× PBS containing 0.2% Triton X-100 (Sigma-Aldrich, St. Louis, MO, USA) was used to block non-specific binding, after which cells were incubated with the primary antibody (anti-PCV2, rabbit polyclonal, 1:500 dilution) diluted in 1× PBS with 1% BSA for 1 h. Cells were washed with 1× PBS three times, for 5 min each time. Next, cells were incubated with the secondary antibody conjugated with Alexa-488 (Invitrogen, Carlsbad, CA, USA) (anti-rabbit, goat polyclonal, 1:1000 dilution) diluted in 1× PBS with 1% BSA for 50 min and washed with 1× PBS three times, for 5 min each time. To indicate the nucleus, cells were stained with DAPI (Invitrogen, Carlsbad, CA, USA) for 5 min and washed with 1× PBS three times, for 5 min each time. For imaging, cells were maintained in 1× PBS, which enabled storage for up to 1 week. To preserve cells for longer intervals, they were washed three times with sterile water to remove salts, air-dried, and sealed for storage at 4 °C.

### 2.6. Confocal Laser Scanning Microscopy

To allow imaging of the subcellular localization of the PCV2 Cap and mutants, 5 × 10^5^ cells were placed into a 35 mm glass-bottomed Petri dish and transfected with 2 μg plasmid. At 24 h after transfection, confocal images were recorded using a Live SR spinning disk confocal microscope (Live SR CSU W1, Nikon, Japan) equipped with a sCMOS Prime 95B (Nikon, Japan) and a 100 × 1.4 NA oil objective (Nikon, Japan). To image cells transfected with infectious clone DNA, cells from passage 4 were added to a 35 mm glass-bottomed Petri dish. After cell attachment, cells were fixed, and the indirect immunofluorescence assay (IFA) was performed as described above. Confocal images were recorded with a Live SR spinning disk confocal microscope (Live SR CSU W1, Nikon, Japan) equipped with a sCMOS Prime 95B (Nikon, Japan) and a 10× objective (Nikon, Japan). DAPI, EGFP/Alexa-488, RFP/Alexa-561, and Alexa-647 were excited with 405, 488, 561 and 647 nm lasers, respectively. Images in nd2 format were analyzed with ImageJ (National Institutes of Health, Bethesda, MD, USA).

### 2.7. Flow Cytometry

For detection of apoptosis, cells were transfected with a plasmid (Cap and Cap-DMLS) for 24 h, digested, collected in 1.5 mL tubes and stained with an Annexin-V/PI staining kit (Solarbio, Beijing, China) according to the manufacturer’s instructions. Thereafter, cells were analyzed by flow cytometry (BD FACSVerse, BD Biosciences, San Jose, CA, USA). For each group, 10,000 cells were recorded. Finally, raw data were analyzed with FlowJo software (BD Biosciences, San Jose, CA, USA).

### 2.8. Prediction of the Cellular Localization of Proteins

Prediction analysis was performed using the amino acid (aa) sequences of the PCV2 Cap protein (ID: O56129), thioredoxin (ID: Q99757) and nucleolin 1 (ID: Q9FVQ1). Two internet-based tools, WoLF PSORTII [25] and PSORT II [26], were used to predict the cellular localization of proteins based on their amino acid sequences.

## 3. Results

### 3.1. Prediction of the MLS in PCV2

Two prediction servers (PSORTII [26] and WoLF PSORTII [25]) were used to predict the subcellular location of PCV2 Cap. We first tested the reliability of the two methods with the following two proteins, whose subcellular localization was well known: thioredoxin and nucleolin 1. Thioredoxin is localized to the mitochondria and is important for the control of mitochondrial reactive oxygen species homeostasis, apoptosis regulation, and cell viability. In contrast, nucleolin 1 is localized in the nucleus and is associated with intranuclear chromatin and pro-ribosomal granules. The predicted results were consistent with the established cellular localization of thioredoxin and nucleolin 1 (Table 1).

For PCV2 Cap, PSORTII and WoLF PSORTII clearly predicted cellular localization in the mitochondria (Table 1). Although the nucleus was also identified as a potential location of PCV2 Cap, its probability was lower than that of the mitochondria. The WoLF PSORTII algorithm revealed high similarity between PCV2 and many proteins localized in the mitochondria, including thymidine kinase 2, which phosphorylates thymidine and deoxycytidine in the mitochondrial matrix. The PSORT II algorithm predicted that, for the N-terminal 64 amino acid cleavable signal peptide, the predicted probabilities of mitochondrial localization and nuclear localization are 56.5% and 34.8%, respectively (Figure 1).

### 3.2. The PCV2 Cap Protein (16–42) Region Functions as a Mitochondrial Localization Signal

To confirm the location of the MLS of the PCV2 Cap protein, we first fused the predicted MLS (1–64) with the EGFP Cap(1–64)-EGFP and transfected the resulting fusion protein into PK15 cells to observe its subcellular localization by confocal fluorescence microscopy. Cap(1–64)-EGFP was mainly localized in the nucleus after 24 h of transfection, consistent with Cap(1–41) (Figure 2A,B). By analyzing the distribution of the fluorescent signal, we concluded that Cap(1–64)-EGFP should be trapped in the nucleolus. Considering that the NLS of the Cap protein may affect the function of the MLS, a truncated mutant removal of NLS Cap(42–64)-EGFP was constructed and transfected into PK15 cells. Confocal imaging showed that the subcellular location of Cap(42–64)-EGFP was similar to that of EGFP expressed alone, presenting a diffuse distribution in the nucleus and cytoplasm (Figure 2B). Summarizing these results, we proposed the hypothesis that the NLS of Cap may have dual localization functions. To verify this hypothesis, we constructed two truncated mutants, Cap(1–22)-EGFP and Cap(16–41)-EGFP, based on the structural features of NLS, containing one α-helix and two disordered fragments (Figure 2A). Then, we transfected them into PK15 cells and confirmed their subcellular localization with confocal fluorescence microscopy. Surprisingly, Cap(16–41)-EGFP had a rod-like and granular distribution, and was mainly localized in the cytoplasm 24 h after transfection; this finding was completely distinct from the previously observed nucleolar distribution (Figure 2B). More importantly, Cap(16–41)-EGFP co-localized with the mitochondria, which were identified with TXN2-DsRed (Figure 2D). Cap(1–22)-EGFP exhibited a nuclear distribution, and it was found to be trapped in the nucleolus and co-localized with fibrillarin (FBL) (Figure 2C). These results strongly suggest that the N-terminal Cap (16–41) can function as an MLS. The N-terminus of the Cap protein has dual localization functions; the NLS is the dominant signal, whereas the mitochondrial signal may function at specific periods during viral infection.

### 3.3. The MLS of the PCV2 Cap Plays a Role in Cap-Induced Apoptosis

The PCV2 Cap has been reported to induce mitochondrial apoptosis in PK15 cells. To confirm the role of the MLS in Cap-induced apoptosis, a Cap mutant (Cap-DMLS), with an MLS deletion, was constructed and transfected into PK15 cells. Apoptosis was monitored for 24 h after transfection of the cells with plasmids. The apoptosis rate of the cells transfected with Cap-DMLS (11.7%) was significantly decreased compared to that of the cells transfected with wild-type Cap (16.8%) in PK15 cells, demonstrating that the Cap MLS plays a role in Cap-induced apoptosis (Figure 3A,B).

### 3.4. The MLS of the PCV2 Capsid Protein Is Critical for Viral Propagation

In a previous study, PCV2 was rescued from PK-15 cells transfected with an infectious DNA clone containing two copies of stem loops, but not rescued from cells transfected with a DNA clone carrying only one copy of the stem loop [24]. To determine the function of the MLS, an infectious DNA clone mutant, with deletion of the MLS, was prepared and transfected into PK-15 cells. In the PK-15 cell culture, the viral genome of the Cap-DMLS mutant was not detectable after passage 4, consistent with a negative control containing only one copy of the stem loop, whereas a stable copy number of viral genomic DNA was detected in the positive control containing two copies of stem loops until passage 8 (Figure 4B). Indirect immunofluorescence assays (IFA) for the Cap protein in cells of the fourth generation also revealed that PCV2 was not rescued from the Cap-DMLS mutant (Figure 4A). These results strongly suggested that the MLS deletion of Cap is lethal for virus rescue.

## 4. Discussion

Mitochondria are double membrane-bound organelles present in most eukaryotes; their most important function is to generate energy through oxidative phosphorylation [27]. In addition, mitochondria are involved in a variety of cellular metabolic processes, including proliferation, differentiation, apoptosis, senescence, and calcium homeostasis [27]. Alterations in mitochondrial function and morphology can cause a variety of diseases [28]. Mitochondria have an important role in viral infections [29,30]. Most mitochondrial proteins are localized to the mitochondria by the MLS; for example, apoptosis-inducing factor (AIF) and endonuclease G are involved in apoptosis, and both proteins have an MLS located at the N-terminus [31]. The MLS possess the following several common features: (1) they are mostly located at the N-terminal of the peptide chain and consist of 15–70 amino acids; (2) they have no negatively charged amino acids and form an amphipathic α-helix; (3) there is no specificity requirement for the protein being transported, and non-mitochondrial proteins attached to such signal sequences will also be transported to the mitochondria. PCV2 infection has been shown to lead to apoptosis, although the mechanism underlying this effect is not fully understood [32]. PCV2 Cap, but not PCV2 Rep, induced caspase-3 cleavage, indicating that PCV2 induced apoptosis [15]. The ORF3 protein of PCV2 induced apoptosis alone, involving the caspase-8 pathway [33]. More importantly, the Cap also induced ORF3-independent mitochondrial apoptosis via PERK activation and elevation of cytosolic calcium concentrations [34]. Previous studies have shown that the N-terminal of PCV2 Cap was highly conserved [35], contains an α-helix (16–22), and is rich in positively charged basic residues [36]. In this study, the subcellular location of PCV2 Cap was predicted, and the results showed that the PCV2 Cap was found to have high similarity with many mitochondrial proteins, and the probability of PCV2 Cap localization to the mitochondria was 56.5% over 34.8% for nuclear localization, suggesting that the PCV2 Cap may have an MLS. Confocal fluorescence images clearly demonstrated that PCV2 Cap(16–41) co-localized perfectly with mitochondrial markers, suggesting that it functions as an MLS (Figure 2B,D). More importantly, removal of the MLS attenuated Cap-induced apoptosis (Figure 3B). This is known; the novelty is that the MLS sequence contributes to apoptosis. In addition, Cap(1–22) could perform the functions of the NLS, similarly to Cap(1–41) (Figure 2B), suggesting that the NLS of PCV2 Cap requires redefinition and the mechanisms of nuclear transport need to be further investigated.

The replication site of PCV2 has been controversial. PCV2 replicates in the nucleus in mitotic cells via a process dependent on host cellular DNA polymerase. In contrast, in non-dividing macrophages, PCV2 may replicate in the mitochondria in a specific manner [37,38], as indicated by the presence of aggregated immature viral particles in mitochondria [21,22]. Exploration of the ultrastructure of PCV2-infected cells revealed ICIs, composed of PCV2 VLPs, surrounding and filling proliferating and severely swollen mitochondria [22]. In this study, we determined that the MLS of the PCV2 Cap is contained in its NLS, indicating that the process of PCV2 Cap protein subcellular localization is complex, and suggesting the possibility that PCV2 shuttles between the mitochondria and the nucleus in specific cell types, at certain times during infection. Furthermore, we investigated the relationship between the PCV2 Cap MLS and virus replication. We attempted to rescue the virus from cells infected with clone DNA missing the MLS; however, deletion of the MLS resulted in the failure of virus rescue (Figure 4A,B), suggesting that MLS deletion of Cap is lethal for virus rescue. Therefore, the relationship between the PCV2 MLS and viral replication in primary cells requires further confirmation.

In summary, the MLS of the PCV2 Cap protein was identified for the first time, and it was found to play an essential role in virus rescue in PK-15. These findings provide insights into the mechanisms underlying the subcellular localization and replication of circovirus, as well as a new target for antiviral compounds.

## Figures and Tables

**Figure 1 vetsci-08-00272-f001:**
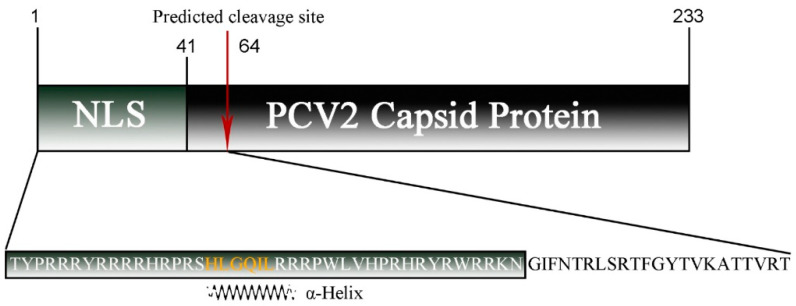
Schematic diagram of the PCV2 capsid protein sequence with the NLS region and predicted cleavage site (red arrow) indicated. The sequence of the predicted MLS used in this study is shown below, where an α-helix is marked in orange.

**Figure 2 vetsci-08-00272-f002:**
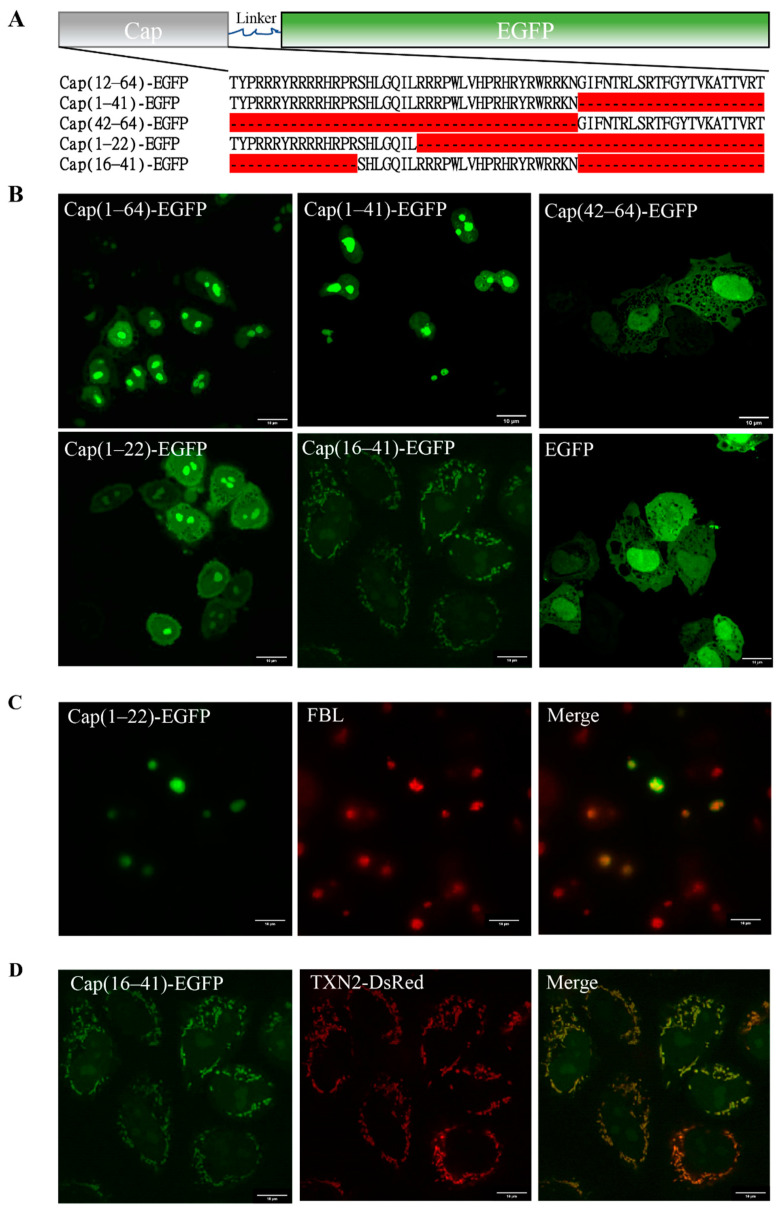
Subcellular localization of PCV2 Cap and mutations. (**A**) The sequence information of recombinant protein Cap-EGFP and mutations. (**B**) Confocal images of the over-expressed Cap(1–64)-EGFP, Cap(1–41)-EGFP, Cap(42–64)-EGFP, Cap(1–22)-EGFP, Cap(16–41)-EGFP and EGFP in PK15 cells. EGFP signal was indicated in green in PK15 cells. Scale bar: 10 μm. (**C**) Co-localization of Cap(1–22)-EGFP (green) and FBL (red) in PK15 cells. (**D**) Co-localization of Cap(16–41)-EGFP (green) and TXN2-DsRed (red) in PK15 cells.

**Figure 3 vetsci-08-00272-f003:**
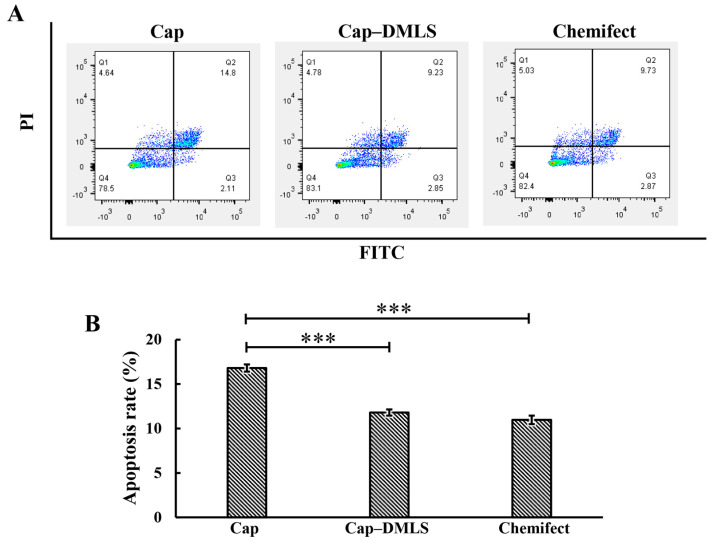
Apoptosis of PCV2 Cap and Cap-DMLS in PK15 cells was analyzed by flow cytometry. PK-15 cells were transfected with recombinant plasmids of Cap and Cap–DMLS for 24 h. Precipitated cells were acquired by centrifugation, stained with Annexin-V/ propidium iodide (PI) staining kit and detected using flow cytometry. (**A**) Scatter plot of representative flow cytometry of Annexin-FITC (X-axis) and propidium iodide (PI) (Y-axis). Cells expressing Cap (left panel), cells expressing Cap-DMLS (middle panel), and cells with no DNA transfected as control (right panel). (**B**) Bar graph presented the average quantitative results of three independent flow cytometry experiments in PK15 cells (*** *p* < 0.001).

**Figure 4 vetsci-08-00272-f004:**
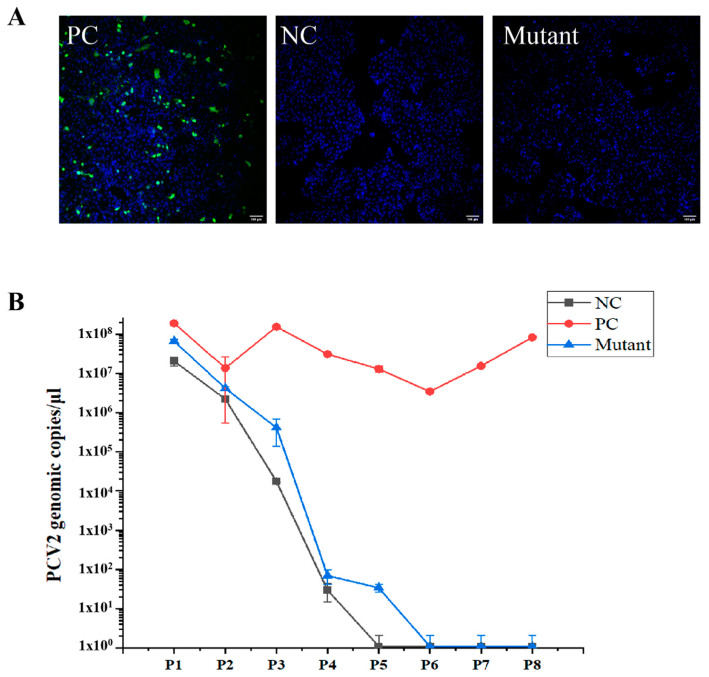
PCV2 wild-type virus and mutant infectious DNA clone were rescued. PK15 cells were transfected with various PCV2 infectious clone DNA (WT and DMLS mutant), and then the cells grew for more than 8 passages. (**A**) The confocal images of PK15 cells transfected with various PCV2 infectious clone DNA of WT (positive control, PC) and mutant, and immunofluorescence staining of PCV2 Cap. The infectious clone DNA containing only one copy of stem loop was transfected as negative control (NC). Green signal indicated the Cap and blue signal indicated the nucleus. Scale bar: 100 μm. (**B**) PCV2 genomic copies in each passage were detected by quantitative real-time PCR. The horizontal axis represented the numbers of cell passages, whereas the vertical axis represented the viral genomic copies. The results of three independent experiments were used to analyze.

**Table 1 vetsci-08-00272-t001:** Prediction of the subcellular localization of proteins.

*PCV2 Cap*	PSORII (%)	WoLF PSORTII (%)
Mitochondrion	56.5	15.0
Nucleus	34.8	14.0
Cytoplasm	8.7	3.0
** *Thioredoxin* **		
Mitochondrion	95.0	31.0
Nucleus	4.3	0
Cytoplasm	0	0
** *Nucleolin 1* **		
Mitochondrion	0	0
Nucleus	91.3	32.0
Cytoplasm	4.3	0

## Data Availability

The data presented in this study are available in the manuscript.

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
