# Peer review of "Mitochondrial Localization Signal of Porcine Circovirus Type 2 Capsid Protein Plays a Critical Role in Cap-Induced Apoptosis"

_vetsci, 2021, doi:10.3390/vetsci8110272_

Round 1
Reviewer 1 Report
Infection with PCV2, the causative agent of post-weaning multi-systemic wasting syndrome (PMWS) in pigs, has been associated with histopathological changes in lymphoid tissues, including presence of PCV2-like particles (VLPs) in mitochondria of lymphocytes and macrophages, suggesting that mitochondria are involved in PCV2 life cycle. Here, Yu et al identified a mitochondrial localization signal (MLS) in the porcine circovirus 2 (PCV2) capsid protein (Cap). Deletion of MLS sequences from Cap was associated with attenuation of Cap-induced apoptosis, and an infectious clone lacking the MLS was unable to propagate after transfection of cells. The manuscript is well written and figures appropriate. Whether the identified MLS plays a role in directing Cap to the mitochondria during virus infection is still an open question. Also no light is shed on the mechanisms allowing Cap to use two sorting signals or if signals are cell-type specific. Issues regarding clarity and presentation are commented below.
1- P4, L158. PSORT II predicts no signal peptide for Cap, please clarify. Also the authors should indicate whether one or more Cap sequences were examined. For example, Cap from strain HN-YY-1-201903 is predicted nuclear over mitochondrial (39% vs 30%).
L159, is NLS at position 41 predicted or previously determined experimentally? PSORT II predicts no pat/bipartite NLS for position 41 of Cap. Clarify.
2- Fig 2. Lack of DAPI counterstaining complicates interpretation of label localization. From L181 it is assumed that the bright areas in Fig 2B represent nucleoli. If so, this should be described in L168.
Were PK15 cells tested for persistent PCV1? If so, state it in M&M section.
3- L174 and 186. What the authors mean with “a dual-signal peptide function”? The terms signal peptide usually refer to the N-terminal peptide of proteins destined to the secretory pathway. Clarify.
L185. Perhaps “that the N-terminal Cap (16-41) can function as a MLS” better describes the
findings.
4- L217, ref 24 is likely not the correct citation for PCV2 infectious clones. M. Fenaux, P. G. Halbur et al, 2002, seems to be the indicated ref. L218, Is the mutant MLS infectious clone a two tandem copy too? If so, state it clearly in L218. Do the authors have an explanation for this two vs one copy effect for their clones?
5- L220, after passage 5 according to fig. 4B.
Did the authors attempt to recover infectious virus after transfection with the mutant clone? If so, was the preparation titrated and compared to virus harvested after transfection with wild type infectious clone? They should explain why they chose to test viral propagation by the more indirect transfection /PCR approach.
6- Fig 4, Indicate in legend the number of times experiment in fig 4B was repeated.
Spell NC and PC.
7- L1-3. The title is misleading as no viral pathogenesis studies are described in the manuscript.
8- L255-256, this is known; the novelty is that MLV sequence contribute to apoptosis.
9- L270-271, the statement is awkward as no macrophages were used in these experiments.
Author Response
Dear Dr.,
Thanks for your reviewing our articles and we really appreciated the constructive comments from you. As requested, we have revised this manuscript and carefully checked each point. Necessary changes made to the text were highlighted in red.
Point 1: P4, L158. PSORT II predicts no signal peptide for Cap, please clarify. Also the authors should indicate whether one or more Cap sequences were examined. For example, Cap from strain HN-YY-1-201903 is predicted nuclear over mitochondrial (39% vs 30%).
Reply: Thanks for your comments. We apologize for failing to describe clearly. The PSORT II algorithm predicted that the probabilities of mitochondrial localization and nuclear localization are 56.5% and 34.8%, respectively. We have revised this part and may be easier for our readers to understand the meaning, please see line 159-161. Indeed, we just showed one Cap (ID: O56129, we have described in Materials & Methods, please see line 140) prediction, but we tested several Cap sequences and obtained similar results, probably because the N-terminal of Cap is highly conserved (Ref 35). Here, our predicted results of Cap from strain HN-YY-1-201903 is mitochondrial over nuclear (60.9% vs 30.4%).
Point 2: L159, is NLS at position 41 predicted or previously determined experimentally? PSORT II predicts no pat/bipartite NLS for position 41 of Cap. Clarif.
Reply: Thanks for your comments. NLS (1-41) was determined by Liu Qiang (Ref 12) in 2001. In our results, we found Cap (1-22) could perform the function of the NLS just like Cap (1-41)(Fig 2B). We also found PSORT II predicts no pat/bipartite NLS for position 41, and all the predicted pat/bipartite located Cap(1-25). (pat4: PRRR (4) at 4, pat4: RRRR (5) at 9, pat4: RRRH (3) at10, pat4: RRHR (3) at 11, pat4: RRRP (4) at 24, pat7: PRRRYRR (5) at 4, bipartite: RRRHRPRSHLGQILRRR at 10, bipartite: RRHRPRSHLGQILRRRP at 11, bipartite: RRRPWLLHPRHRYRWRR at 24, bipartite: RRPWLLHPRHRYRWRRK at 25)
Point 3: Fig 2. Lack of DAPI counterstaining complicates interpretation of label localization. From L181 it is assumed that the bright areas in Fig 2B represent nucleoli. If so, this should be described in L168.
Reply: Thanks for your comments. We have made necessary changes, please see line 172-173.
Point 4: Were PK15 cells tested for persistent PCV1? If so, state it in M&M section.
Reply: Thanks for your comments. The PK15 cell line used in our study is free of PCV1. We added the necessary information it in M&M section, please see line 70.
Point 5: L174 and 186. What the authors mean with “a dual-signal peptide function”? The terms signal peptide usually refer to the N-terminal peptide of proteins destined to the secretory pathway.Clarify.
Reply: Thanks for the suggestion. Here “a dual-signal peptide function” means PCV2 Cap owns the functions of NLS and MLS. We have changed it with “dual localization functions”, please see line 179 and 190.
Point 6: L185. Perhaps “that the N-terminal Cap (16-41) can function as a MLS” better describes the findings.
Reply: Thanks for the suggestion. We have corrected it, please see line 189.
Point 7: L217, ref 24 is likely not the correct citation for PCV2 infectious clones. M. Fenaux, P. G. Halbur et al, 2002, seems to be the indicated ref. L218, Is the mutant MLS infectious clone a two tandem copy too? If so, state it clearly in L218. Do the authors have an explanation for this two vs one copy effect for their clones?
Reply: Thanks for your comments. In fact, we have constructed an infectious DNA clone containing 1.1 copy of the PCV2 genomic DNA with two stem loops (Fig. 7B in ref 24.) instead of two tandem copies of the viral genome, and we confirmed that only the WT with two copies of stem loop (served as PC in this study) can rescue PCV2 while WT with one copy of stem loop (served as NC in this study) could not rescue PCV2 in previous study. We have corrected this part and may be easier for our readers to understand the processes. Please see the lines 220-222 in the revised version.
Point 8: L220, after passage 5 according to fig. 4B.
Did the authors attempt to recover infectious virus after transfection with the mutant clone? If so, was the preparation titrated and compared to virus harvested after transfection with wild type infectious clone? They should explain why they chose to test viral propagation by the more indirect transfection /PCR approach.
Reply: Thanks for your comments. Indeed, we attempt to rescued PCV2 from PK15 cells after transfection with the mutant clone through indirect immunofluorescence assays (Fig. 4A), and compared the copies of viral genome from PK15 cells transfected with wild type DNA clone or mutant clone by PCR approach (Fig. 4B). If the PCV2 DNA copy numbers gradually decreased with continuous passages in the cell culture and lower than that of the wild type, it suggested the mutant plays critical roles in virus rescue and propagation in PK15 cell culture. We have revised this part and may be easier for our readers to understand the meaning. Please see the lines 222-231 in the revised version.
Point 9: Fig 4, Indicate in legend the number of times experiment in fig 4B was repeated. Spell NC and PC.
Reply: Thanks for your comments. The results of three independent experiments were used to analyze. And The confocal images of PK15 cells transfected with various PCV2 infectious clone DNA of WT (positive control, PC) and Mutant and immunofluorescence stainning of PCV2 Cap. The pSP72 vector plasmid was transfected as negative control (NC). We have corrected this part and may be easier for our readers to understand the meaning, please see line 237-242.
Point 10: L1-3. The title is misleading as no viral pathogenesis studies are described in the manuscript.
Reply: Thanks for your comments. We have changed the title to “Mitochondrial localization signal of porcine circovirus type 2 capsid protein is critical for Cap-induced apoptosis and virus replication”, please see line 1-3.
Point 11: L255-256, this is known; the novelty is that MLS sequence contribute to apoptosis.
Reply:Thanks for your comments. We have corrected it, please see line 271-272.
Point 12: L270-271, the statement is awkward as no macrophages were used in these experiments.
Reply:Thanks for your comments. Here, we would like to state that we will use primary cells for experiments in the future, and we have corrected it, please see line 289.
Reviewer 2 Report
The manuscript describes for the first time the identification of a mitochondrial localization signal (MLS) in the N-terminus of the Cap protein of PCV2. Firstly, the authors used two servers that predicted nuclear and mitochondrial cellular localization for Cap. Then, fusions of different regions of the first 64 aa to a reporter protein and the results showed that the sequence 1-22 was responsible for nuclear localization of EGFP, whereas the sequence 16-41 was responsible for mitochondrial localization of EGFP in confocal fluorescence microscopy experiments. Colocalization assays with TXN2-DsRed and fibrillarin confirmed these findings. Finally, the authors showed that the deletion of MLS affects the apoptosis of MLS.
The manuscript has several major concerns:
- In Cap-DMLS the authors did not specify the precise deletion. Is it 16-41? If that construction includes aa from the nuclear localization signal, why do the authors state that MLS is responsible for apoptosis? (line 83).
- No experiments support the statement that NLS is the dominant signal, whereas the mitochondrial signal may function at specific periods during viral infection.
- The author refers to position 64 as a cleavage site. Do the authors suppose that cleavage occurs as a necessary process in natural infection?
- It is not clear why they designed the CAP(16-41)-eGFP ending at position 41 as the predicted NLS if the predicted MLS ended at position 64. Moreover, it would be interesting to add in the discussion why they think the server predicted the complete sequence 1-64 as the MLS.
Minor concerns:
- Line 31 Correct to “approximately 30 years ago”.
- Line 93 Specify what is a suitable time.
- Line 100 What primer did they use?
- Line 108 Change “modified” for “conjugated”
- Line 122 Define IFA the first time the abbreviation is used.
- Lines 157-159 Please, clarify in the manuscript what do the authors mean with “the NLS at position 41 of this peptide”, is it 1-41?
- Line 177 What do the authors mean with “disordered fragments”? Please, clarify this point.
- Figure 2: The figure could be improved by showing the bright field of the pictures.
- Figure 1 and Figure 3 look horizontally stretched. Please, correct them.
- Line 207 Define propidium iodide (PI) the first time.
- Line 215 PK15 is different from the rest of the text (PK-15). Please, uniform this point along the manuscript.
- Figure 4 Please indicate in the caption what the abbreviations PC and NC mean.
- Line 244 Use the abbreviation MLS
- Line 250 In the discussion, the authors note that “In this study, the PCV2 Cap was found to have high similarity with many mitochondrial proteins, e.g., thymidine kinase 2”. Nevertheless, it is not clear what they refer to with this sentence.
- Line 253 The authors referred to the CAP(16-41) as NT(17-41). Please, clarify this point.
Author Response
Dear Dr.
Thanks for your reviewing our articles and we really appreciated the constructive comments from you. Followed these suggestions, we tried our best to revise the manuscript. Necessary changes made to the text were highlighted in red.
The manuscript has several major concerns:
1. In Cap-DMLS the authors did not specify the precise deletion. Is it 16-41? If that construction includes aa from the nuclear localization signal, why do the authors state that MLS is responsible for apoptosis? (line 83).
Reply: The suggestion is quite relevant to make the manuscript clearer. Yes, Cap-DMLS represents the precise deletion of 16-41. MLS sequence located in 16-41 aa of Cap. We have marked and rewritten this for easier understanding and please see line 84. In 2001, Liu Qiang et, al have reported that the NLS of PCV2 Cap was the N-terminal 41 amino acids (Ref 12). Here, we found Cap (1-22) could perform the function of the NLS (Fig 2B), interestingly we found Cap (16-41) can function as an MLS (Fig 2B and 2D), further deletion of the MLS (16-41) attenuated Cap-induced apoptosis (Fig 3), so we state that MLS is responsible for apoptosis. We have revised this part and highlighted in the discussion (please see line 268-274), and may be easier for our readers to understand the meaning.
2. No experiments support the statement that NLS is the dominant signal, whereas the mitochondrial signal may function at specific periods during viral infection.
Reply: Thanks for your comments. As the sole structural protein of PCV2, accumulation studies have been reported on the subcellular localization of PCV2 Cap. The following points are widely recognized: 1) expression of PCV2 Cap was almost located in the nucleus of cell lines; 2) a special arginine-rich N-terminus (position 1-41) of PCV2 Cap is described as a NLS for its accumulation into the nucleus, and our results of Fig 2B are consistent with this; 3) The replication site of PCV2 was in the nucleus, and PCV2 Cap was involved in the replication of the viral genome as well as the assembly of the virus; 4) electron microscopy studies have shown that during viral infection of the host, PCV2 Cap was distributed in both the nucleus and the mitochondria. All evidences suggested that the main behavior of Cap is performed in the nucleus through the NLS. However, there are no reports on how PCV2 Cap localized to the mitochondria. In this work, we confirmed Cap (16-41) can function as an MLS, but it is not clear how it is activated during viral infection. We are now trying to figure out the key factors involved in the switch of Cap from nucleus to mitochondria.
3. The author refers to position 64 as a cleavage site. Do the authors suppose that cleavage occurs as a necessary process in natural infection?
Reply: Thanks for your comments. Many mitochondrial proteins localize to the mitochondria and then undergo cleavage. As the simplest virus infected the eukaryotic, Cap is the only structural protein of PCV2. Obviously, it is not a functionally simple mitochondrial-associated protein, it is involved in many viral life processes. More importantly, intact virus-like particles have been observed in mitochondria. Therefore, we think cleavage occurs is not a necessary process in natural infection.
4. It is not clear why they designed the CAP(16-41)-eGFP ending at position 41 as the predicted NLS if the predicted MLS ended at position 64. Moreover, it would be interesting to add in the discussion why they think the server predicted the complete sequence 1-64 as the MLS.
Reply: Thanks for your comments. In fact, at the beginning we thought the MLS was 42-64, because 1-41 had been identified as the NLS. We fused Cap (42-64) and EGFP, but the distribution of Cap (42-64)-EGFP was the same with expression of EGFP alone (Fig 2B). Therefore, we considered whether there would be an overlap between the NLS and the MLS. Based on secondary structure analysis, the NLS of the PCV2 Cap contains an α-helix (16-22) (Fig 1), and 1-16 and 23-41 are disordered fragments, which indicates that they do not have a fixed secondary structure. Here, we designed the mutations with the α-helix preserved. Mitochondrial predictions are based mainly on the common features of existing MLS. 1)They are mostly located at the N-terminal end of the peptide chain and consisting of 15-70 amino acids. 2) They have no negatively charged amino acids and form an amphipathic α-helix. 3) There is no specificity requirement for the protein being transported, and non-mitochondrial proteins attached to such signal sequences will also be transported to the mitochondria. PCV2 Cap(1-64) contains an α-helix (16-22) and is rich in positively charged basic residues (Fig 1). We have revised this part and make the manuscript clearer, please see line 252-257.
Minor concerns:
- Line 31 Correct to “approximately 30 years ago”.
Reply: Thanks for your comments. We've corrected it and please see line 32.
- Line 93 Specify what is a suitable time.
Reply: Thanks for your comments. We've corrected it and please see line 94.
- Line 100 What primer did they use?
Reply: Thanks for your comments. Primer has been added please see line 101-102.
pPCV2-cap1-qpcr-F (ctgttttcgaacgcagtgcc)
pPCV2-cap1-qpcr-R (aactactcctcccgccatac)
- Line 108 Change “modified” for “conjugated”
Reply: Thanks for your comments. We've corrected it and please see line 110.
- Line 122 Define IFA the first time the abbreviation is used.
Reply: Thanks for your comments. We've corrected it and please see line 124-125.
- Lines 157-159 Please, clarify in the manuscript what do the authors mean with “the NLS at position 41 of this peptide”, is it 1-41?
Reply: Thanks for your comments. Yes, it is 1-41, we have rewritten this part and please see line 159-161.
- Line 177 What do the authors mean with “disordered fragments”? Please, clarify this point.
Reply: Thanks for your comments. Based on secondary structure analysis, the NLS of the PCV2 Cap was composed of two stretches (disordered fragments) separated by an α-helix (Fig 1). The disordered fragment indicates that it does not have a fixed secondary structure. Here, to determine the sequence of the MLS, we designed the mutations with the a-helix preserved.
- Figure 2: The figure could be improved by showing the bright field of the pictures.
Reply: Thanks for your comments. Indeed, bright field images make it easy to observe the morphology of the cells, but we did not acquire high-quality bright field images. Here, we used spinning-disk confocal imaging to capture the middle layer of cells with a thickness of 0.2 µm, combined with co-localization with organelles (mitochondria and nucleolus) to show the subcellular distribution of Cap-EGFP and mutants.
- Figure 1 and Figure 3 look horizontally stretched. Please, correct them.
Reply: Thanks for your comments. We've corrected it and please see line 163 and 208.
- Line 207 Define propidium iodide (PI) the first time.
Reply: Thanks for your comments. We've corrected it and please see line 212.
- Line 215 PK15 is different from the rest of the text (PK-15). Please, uniform this point along the manuscript.
Reply: Thanks for your comments. We've corrected it and please.
- Figure 4 Please indicate in the caption what the abbreviations PC and NC mean.
Reply: Thanks for your comments. We've corrected it and please see line 237-238.
- Line 244 Use the abbreviation MLS
Reply: Thanks for your comments. We've corrected it and please see 252.
- Line 250 In the discussion, the authors note that “In this study, the PCV2 Cap was found to have high similarity with many mitochondrial proteins, e.g., thymidine kinase 2”. Nevertheless, it is not clear what they refer to with this sentence.
Reply: Thanks for your comments. Here, we were going to describing PCV2 Cap having the characteristics of a mitochondrial protein, and together with major concern 4, we have modified this part and may be easier for our readers to understand the meaning and please see line 252-268.
- Line 253 The authors referred to the CAP(16-41) as NT(17-41). Please, clarify this point.
Reply: Thanks for your comments. It's a mistake in writing, and we've corrected it and please see line 269.